# Neopterin and CXCL-10 in Cerebrospinal Fluid as Potential Biomarkers of Neuroinvasive Dengue and Chikungunya

**DOI:** 10.3390/pathogens10121626

**Published:** 2021-12-15

**Authors:** Marzia Puccioni-Sohler, Samya J. da Silva, Luiz C. S. Faria, David C. B. I. Cabral, Mauro J. Cabral-Castro

**Affiliations:** 1Post-Graduation Programme in Infectious and Parasitic Diseases, Universidade Federal do Rio de Janeiro, Rio de Janeiro 21941-913, Brazil; samyajezinesilva@gmail.com; 2Cerebrospinal Fluid Laboratory, Clementino Fraga Filho University Hospital, Universidade Federal do Rio de Janeiro, Rio de Janeiro 21941-913, Brazil; luizclaudio@hucff.ufrj.br (L.C.S.F.); maurojorge@micro.ufrj.br (M.J.C.-C.); 3School of Medicine and Surgery, Universidade Federal do Estado do Rio de Janeiro, Rio de Janeiro 20271-062, Brazil; 4Faculty of Medicine, Universidade Federal do Rio de Janeiro, Rio de Janeiro 21942-959, Brazil; dcuricabral@gmail.com; 5Paulo de Góes Institute of Microbiology, Universidade Federal do Rio de Janeiro, Rio de Janeiro 21941-902, Brazil

**Keywords:** cerebrospinal fluid, inflammatory biomarker, neopterin, CXCL-10, dengue virus, chikungunya virus

## Abstract

Dengue (DENV) and chikungunya viruses (CHIKV) cause severe neurological complications, sometimes undiagnosed. Therefore, the use of more accessible neuroinflammatory biomarkers can be advantageous considering their diagnostic and prognostic potential for aggravated clinical outcomes. In this study, we aimed to evaluate neopterin and C-X-C motif chemokine ligand 10 (CXCL-10) in cerebrospinal fluid (CSF) for the diagnosis of neuroinvasive DENV and CHIKV. We analyzed the CSF of 66 patients with neurological disorders, comprising 12 neuroinvasive DENV/CHIKV, 20 inflammatory control (viral, bacterial, and fungal meningitis, and autoimmune disorders), and 24 noninflammatory control (cerebrovascular disease, dementia, neoplasm). There was no difference between the concentration of CSF neopterin in the neuroinvasive DENV/CHIKV and control groups. However, there was a significant difference in the CXCL-10 level when comparing the neuroinvasive DENV/CHIKV group and the non-inflammatory control (*p* < 0.05). Furthermore, we found a linear correlation between neopterin and CXCL-10 CSF levels in the three groups. For the DENV/CHIKV neuroinvasive diagnosis, the ROC curve showed the best cut-off values for CSF neopterin at 11.23 nmol/L (sensitivity of 67% and specificity of 63%), and for CSF CXCL-10 at 156.5 pg/mL (91.7% sensitivity and specificity). These results show that CXCL-10 in CSF represents an accurate neuroinflammatory biomarker that may contribute to neuroinvasive DENV/CHIKV diagnosis.

## 1. Introduction

The chikungunya (CHIKV) and dengue virus (DENV) are arboviruses transmitted to humans through the bites of *Aedes* mosquito. The DENV virus belongs to the *Flavivirus* genus and can be classified into four serotypes: DENV-1, DENV-2, DENV-3, and DENV-4 [1,2]. The Chikungunya virus belongs to the *Alphavirus* genus and the *Togaviridae* family [3]. Both can cause severe neurological complications, such as encephalitis, encephalopathy, meningitis, myelitis, and Guillain–Barré syndrome. Neurological conditions caused by DENV and CHIKV may be due to direct viral action, local inflammatory reaction, or metabolic disturbances [4,5].

CHIKV and DENV mainly affect human macrophages and monocytes, which secrete cytokines. CXCL-10 is a low molecular weight cytokine involved in immunity and inflammation processes [6,7]. It plays a major role in recruiting T lymphocytes and natural killer (NK) cells during neuroinflammatory reactions. CXCL-10 is expressed in neurons, glial, and stromal cells. Among the CXC family chemokines, CXCL-10 is the most important in inducing inflammatory response by leukocytes in several diseases of the nervous system [8,9].

Neopterin is another important inflammatory biomarker that indicates the activation of the immune system [10]. Some studies have found high cerebrospinal fluid (CSF) levels of neopterin in cases of brain trauma or infections in the meninges, suggesting a local production [11]. Neopterin is independently produced in the brain, and there is no correlation between the concentration of this substance in plasma/serum and CSF [12,13]. Interferon-gamma stimulates cells such as microglia and astrocytes to produce neopterin [13,14]. Biogenic amine-producing cells through the activation of tetrahydrobiopterin (BH4) synthesize neopterin in the hippocampus, amygdala, locus ceruleus, substantia nigra, raphe nuclei, nucleus accumbens, caudate, and putamen. Peripheral neurons of the dorsal ganglia may produce neopterin in inflammatory conditions [13,15]. 

The analysis of these inflammatory biomarkers, such as neopterin and CXCL-10, in the blood allows us to predict the severity of the course of dengue fever and chikungunya. It has been demonstrated that high concentrations of neopterin in the early stages of dengue fever indicated the severity of the disease course [16]. Another study showed a positive correlation between CXCL-10 and anti-CHIKV IgM in the acute phase of the disease, and a positive correlation with anti-CHIKV IgG in the subacute phase [17].

The detection of inflammatory biomarkers in the CSF can provide valuable information about the immune response in the neuroinvasive arboviruses DENV and CHIKV [10]. In addition, neurological symptoms are not specific for neuroarboviruses, and the routine CSF analysis is frequently normal, making diagnosis difficult [18,19]. This study aims to evaluate neopterin and CXCL-10 biomarkers in CSF samples for the diagnosis of neuroinvasive DENV and CHIKV. According to the results found, the analysis of the inflammatory biomarker CXCL-10 in CSF samples can contribute to the early diagnosis and better understanding of the neuropathogenesis of the DENV and CHIKV arbovirus infection.

## 2. Results

Routine CSF analysis was performed in all three study groups. In the group of neurological manifestations associated with neuroinvasive arboviruses, we observed that 6 out of the 12 patients were confirmed with CHIKV infections, and the other 6 had DENV infections. These confirmations were performed by detecting the viral nucleic acid by RT-PCR or specific IgM antibodies in CSF and/or serum as recommended by the CDC guideline (2015) [4]. The neurological complications associated with arbovirus infection consisted of encephalitis (6), Guillain–Barré syndrome (2), myelitis (2), NMOSD (1), and another polyneuropathy (1). The assay of the 12 CSF samples from arbovirus group detected pleocytosis (>4 cells/mm^3^) in seven (58.3%) patients and hyperproteinorrachia (>40 mg/dL) in eight cases (66.7%). All patients had negative results for bacteria, fungi, mycobacteria, and syphilis.

We did not find any difference between age and sex in the three groups (Table 1). The CSF analysis showed a difference in the cell count, but not in the glucose concentration in the groups. The CSF protein and CXCL-10 levels were similar between the arbovirus group and inflammatory control (Table 1 and Figure 1). The CSF neopterin in the arbovirus group had no difference in comparison to both controls (Figure 1). There was a correlation (*r* > 0; *p* < 0.05) between the levels of neopterin and CXCL-10 in all groups—the arbovirus group (*r* = 0.7273, *p* = 0.0096), inflammatory control (*r* = 0.6812, *p* = 0.0009), and noninflammatory control (*r* = 0.5313, *p* = 0.0075)—and between cell count and CXCL-10 (*r* = 0.6822, *p* = 0.0009) only in the inflammatory control (Figure 2). The ROC curve shows cut-off values of 11.23 nmol/L for CSF neopterin with the best sensitivity of 67% and specificity of 63%, and 156.5 pg/mL for CXCL-10 with 91.7% sensitivity and specificity for the diagnosis of arbovirus neuroinvasive DENV and CHIKV (Figure 3). In addition, in the DENV/CHIKV neuroinvasive group, we found in 8 (encephalitis (6), myelitis (1), and another polyneuropathy (1)) of the 12 patients (66.7%) a CSF neopterin concentration higher than 11.23 nmol/L and in 11 (encephalitis (7), myelitis (1), NMOSD (1), GBS (1), and another polyneuropathy (1)) (92%) patients of this group we identified a CSF CXCL-10 concentration higher than 156.5 pg/mL. All cases of neuroarbovirus were in the symptomatic and acute phase of the disease, except for one of CHIKV encephalitis, with symptoms remaining three months after the onset of the disease [20]. This was the only case with CXCL-10 in CSF values below the cut-off in the ROC curve. 

## 3. Discussion

The DENV and CHIKV arboviruses have a high incidence and mortality rate worldwide. Cases of arboviruses that develop into their severe form can affect the nervous system and cause irreparable damage. The detection of inflammatory biomarkers in the cerebrospinal fluid can be extremely important for the early diagnosis and analysis of the evolution of neurological complications. The serological investigation of inflammatory markers CXCL-10 and neopterin provides support for the diagnosis and prognosis of dengue and chikungunya. These biomarkers can indicate whether the virus will progress to a more severe form or remains with mild symptoms. However, most biomarker studies in arboviruses are performed with serological samples. It is important to implement the analysis of these substances also in CSF samples since neopterin and CXCL-10 are inflammatory markers expressed in several nerve cells (glial cells, astrocytes, microglia) [13,14,21]. Both markers are relevant in the function of expressing the evolution and/or intensity of a neuroinflammatory disease of infectious origin. In our study, we analyzed neopterin and CXCL-10 markers in the CSF of patients with neuroinvasive DENV and CHIKV, compared to the inflammatory and non-inflammatory control group. Of all the inflammatory markers (pleocytosis, hyperproteinorrachia, neopterin, and CXCL-10) analyzed in the CSF samples, the protein and CXCL-10 concentration did not show significant differences in the arbovirus groups when compared to the inflammatory control group. The CXCL-10 biomarker in CSF also proved to be an accurate marker to differentiate a neuroinflammatory reaction by DENV and CHIKV in the nervous system in relation to the group of non-inflammatory neurological diseases. There was a positive correlation between CSF neopterin and CXCL-10 markers in all groups, which means that there is a correlation between both inflammatory parameters, which may be locally produced. CXCL-10/IP-10 is an important pro-inflammatory marker in the case of dengue and chikungunya, where there is an increase in this chemokine when induced by IFN-γ, mainly associated with more severe cases and in pregnant women [22,23,24,25,26]. This chemokine attracts T lymphocytes and natural-killer (NK) cells to the site of infection, resulting in faster recovery and the elimination of arboviruses [22]. This process may be aided by competition between CXCL-10 and the virus for the heparan sulfate binding site in cells, preventing viral penetration and replication [26]. In a previous study, there was a significant increase in CXCL-10 in CSF samples in patients with central nervous system (CNS) inflammatory disorders (acute disseminated encephalomyelitis (ADEM), ADEM followed by optic neuritis, anti-N-methyl-D-aspartate receptor encephalitis, Rasmussen encephalitis, acute cerebellitis of unknown etiology, encephalitis of unknown etiology, clinically isolated syndrome, neuromyelitis optica spectrum disorders, and neuroborreliosis) [8]. In cases of tick-borne encephalitis (TBE), the concentration of CXCL-10 was reduced in the period after treatment compared to CSF samples collected in the pre-treatment period, but the concentration of CXCL-10 was still high even after treatment compared to the control group, indicating that CXCL-10 is a good monitoring marker of patients being treated for TBE [27].

In addition, infected monocytes, macrophages, and dendritic cells produce high levels of neopterin, which is a biomarker of the immune system induced by the production of interferon-gamma (IFN-γ) and synthesized by T lymphocytes and NK cells that occurs in arbovirus infection [16,28]. The concentration of neopterin in the initial phase of dengue can serve as an indicator for evolution. The cases of severe dengue had a higher concentration of this biomarker than cases of the mild form of the disease. Its levels increased in infections and decreased after recovery [16,28,29]. This high concentration of neopterin may be associated with an occurrence of more severe inflammation in patients with dengue and chikungunya [16,28,30]. Therefore, the diagnosis of the disease with an emphasis on the early prognosis of the neopterin can improve medical treatment in its initial phase, preventing the progression to its most severe form. Neopterin is expressed in the first days of the disease and the different concentrations of this marker may be useful to monitor the activation of the immune response [16,31]. Another study found higher values of CSF neopterin in patients with meningitis caused by viruses and bacteria compared to controls [13].

In conclusion, we showed a correlation between CSF neopterin and CXCL-10 in inflammatory and non-inflammatory control groups, which means that the concentrations of the two markers are synchronized to increase or decrease, indicating that CXCL-10 and neopterin may have the same mechanism of local production. CSF CXCL-10 proved to be a good diagnostic marker of inflammatory diseases in the CNS, including those caused by the arboviruses DENV and CHIKV. Our findings show that CSF CXCL-10 had a higher sensitivity and specificity for neuroinvasive DENV/CHIKV infection.

## 4. Materials and Methods

### 4.1. Patient’s Samples 

This is a retrospective and cross-sectional study of routine leftover CSF samples of 12 patients with neurological manifestations associated with neuroinvasive arbovirus (DENV/CHIKV), 20 cases with other inflammatory diseases of the central nervous system (CNS) and abnormal routine CSF assay and 24 patients with noninflammatory diseases and normal routine CSF assay. A patient with suspected acute viral infection of the CNS (encephalitis, meningitis, myelitis) or parainfectious/postinfectious syndromes (Guillain–Barré syndrome, another polyneuropathy, neuromyelitis optic spectrum disorder (NMOSD)) was diagnosed with confirmed neuroinvasive arbovirus if the specimens were positive for CHIKV or DENV nucleic acid by reverse transcription PCR (RT-PCR) and/or reactive to specific IgM (ELISA) in CSF [4,32]. The controls tested negative for arbovirus and included an inflammatory group with bacterial/ cryptococcal meningitis (11), other CNS viral infections (encephalitis, meningitis, myelitis) (4), and autoimmune disorders (Guillain–Barré syndrome (GBS), Vogt–Koyanagi–Harada syndrome, optic neuromyelitis) (5). The non-inflammatory group consisted of those with dementia (4), neoplasm (9), psychiatric disturbances (major depression, somatoform-conversion disorder) (2), idiopathic intracranial hypertension (3), metabolic disturbances (6), microangiopathy (1), and degenerative myelopathy (1).

### 4.2. Cerebrospinal Fluid Analysis

For routine CSF analysis, we performed cell count, protein and glucose dosage, direct exam and culture for bacteria, fungi, and mycobacteria, and VDRL tests for syphilis. The neopterin (IBL International GMBH, Hamburg, Germany) and CXCL-10 (Invitrogen Corporation, Camarillo, EUA) biomarkers were evaluated using ELISA commercial kits following the manufacturer’s recommendations.

### 4.3. Statistical Analysis

Statistical analysis was performed using GraphPad software version 9 (GraphPad Software, La Jolla, CA, USA) and Microsoft Excel 2010. Due to the non-Gaussian distribution of data generated by the ELISA tests in the three groups (arbovirus, other inflammatory diseases, and group non-inflammatory), we performed the descriptive statistics based on the median and the first and third quartiles to assess the concentration of neopterin and CXCL-10 markers in the CSF in each sample group. The Kruskal–Wallis non-parametric ANOVA was applied to assess whether there were statistically significant differences in the concentration of markers (neopterin and CXCL-10) between the different groups. For further analysis, Dunn’s multiple comparison tests were used to analyze the specific group pairs if they differed significantly in the concentration of the markers. The ROC curve was applied to calculate the cut-off value of neopterin and CXCL-10 in the two groups (arboviruses and other inflammatory diseases) in relation to the control group (non-inflammatory), based on the sensitivity and specificity values, and Spearman’s test to assess the correlation between inflammatory markers in each group. The differences were considered to be statically significant when *p* < 0.05.

## Figures and Tables

**Figure 1 pathogens-10-01626-f001:**
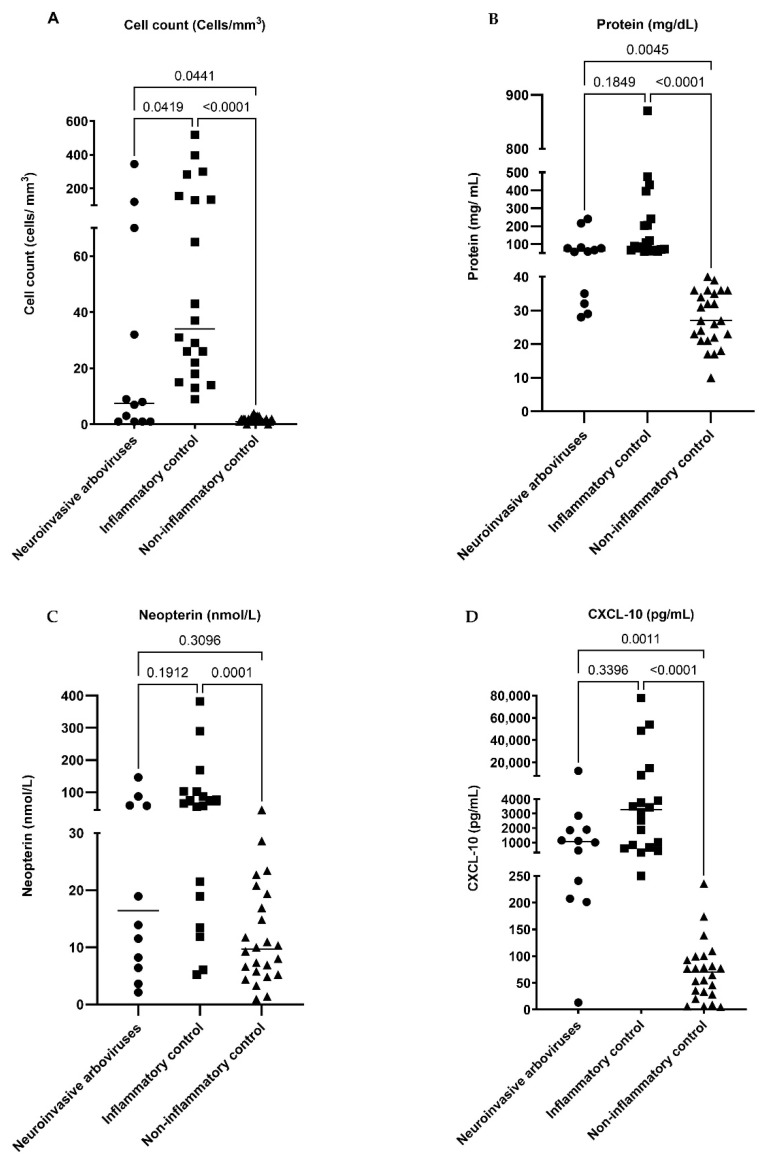
CSF analysis from the different groups of patients (neuroinvasive arboviruses, inflammatory control, non-inflammatory control). The Kruskal–Wallis non-parametric ANOVA was applied to assess whether there were statistically significant differences in the cell count in CSF (**A**), concentration of protein in CSF (**B**), neopterin in CSF (**C**), and CXCL-10 in CSF (**D**) between the groups. For further analysis, Dunn’s multiple comparison tests were used to analyze the specific group pairs. *p* < 0.05: statistical significance.

**Figure 2 pathogens-10-01626-f002:**
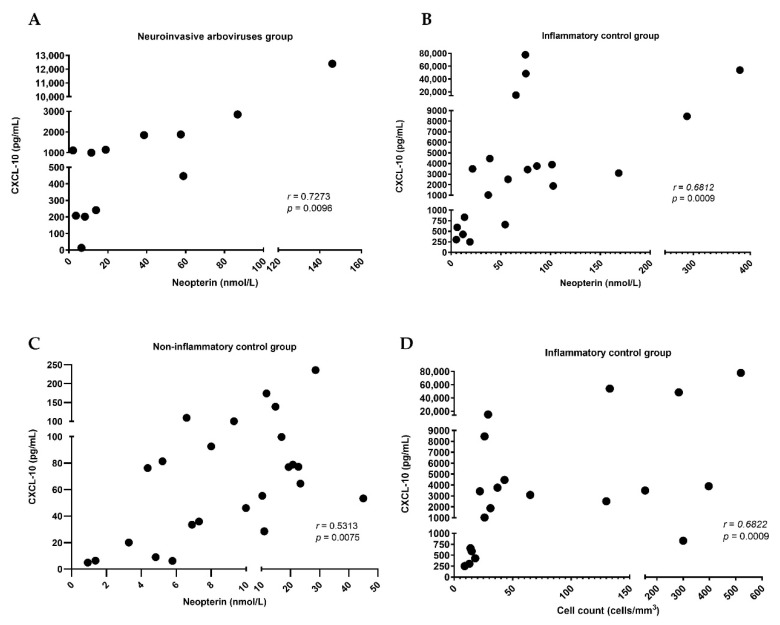
Correlation between CXCL-10 and neopterin in each group and between CXCL-10 and cell count in inflammatory control group: (**A**) neuroinvasive arboviruses group; (**B**) inflammatory group; (**C**) non-inflammatory control group; (**D**) correlation between CXCL-10 and cell count in the inflammatory group. Each dot represents 1 sample. Correlation coefficient (*r*) was calculated using the Spearman test; *r* and *p* values are shown. For the correlation between CXCL-10 and neopterin, the Spearman test demonstrated a very strong correlation (*r* = 0.7273; *p* = 0.0096) in neuroinvasive arboviruses group, and a strong correlation in inflammatory control group (*r* = 0.6812; *p* = 0.0009) and non-inflammatory control group (*r* = 0.5313; *p* = 0.0075). The correlation between CXCL-10 and cell count in neuroinvasive arboviruses group was strong (*r* = 0.6822; *p* = 0.0009). Reference values of Spearman test: *r* ≥ 0.70 is very strong correlation, *r* = 0.40 to 0.69 is a strong correlation, *r* = 0.30 to 0.39 is a moderate correlation, *r* = 0.20 to 0.29 is a weak correlation, and *r* = 0.01 to 0.19 is a negligible correlation or non-correlation. *p* < 0.05 values are significant.

**Figure 3 pathogens-10-01626-f003:**
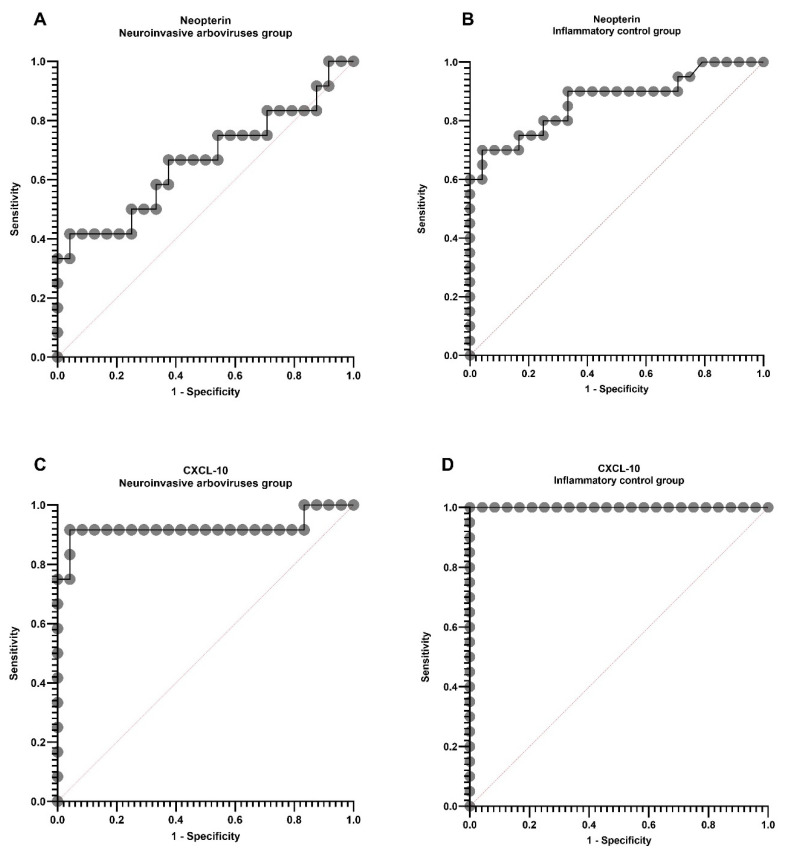
ROC curve comparing neopterin and CXCL-10 in CSF samples. The ROC curve was used to express the non-parametric values found by Dunn’s test, and *p* <0.05 values are significant. The cut-off values that showed the best performance of sensitivity and specificity of the inflammatory markers neopterin and CXCL-10 in CSF samples were selected for the groups of DENV and neuroinvasive CHIKV, inflammatory control, and non-inflammatory control. (**A**) Neopterin in CSF from neuroinvasive arbovirus group versus non-inflammatory control; (**B**) Neopterin in CSF from inflammatory control versus non-inflammatory control; (**C**) CXCL-10 in CSF from neuroinvasive arbovirus group versus non-inflammatory control; (**D**) CXCL-10 in CSF from inflammatory control versus non-inflammatory control.

**Table 1 pathogens-10-01626-t001:** General characteristics of the patients and analysis of the inflammatory marker concentrations in the DENV and CHIKV neuroinvasive group (group A), inflammatory control group (group B), and non-inflammatory control group (group C).

	Group A, DENV and CHIKV Neuroinvasive (*n* = 12)	Group B, Inflammatory Control (*n* = 20)	Group C, Non- Inflammatory Control (*n* = 24)	*p* Value		
				A × B	A × C	B × C
**Sex**						
Female, n (%)	8 (66.7%)	10 (50%)	18 (75%)	0.263	<0.999	>0.999
**Age (years),** median (IQR)	61 (33–64)	37 (30–53.3)	48.5 (31.8–72.5)	0.3485	0.5464	>0.999
**Cell count (cells/mm^3^),** median (IQR)	7.5 (1–60.5)	34 (19–149.5)	1 (1–2)	0.0419 **	0.0441 **	<0.0001 **
**Protein (mg/dL),** median (IQR)	63 (43–79)	86.5 (67–231.3)	27 (21.25–35.75)	0.1849	0.0045 **	<0.0001 **
**Glucose (mg/dL),** median (IQR)	63 (32.8–80)	60 (41.5–80.8)	70.5 (62–87)	>0.999	0.4887	0.1579
**Neopterin (nmol/L),** median (IQR)	16.4 (6.85–58.5)	61.2 (19.5–97.6)	9.6 (5.4–18.7)	0.1912	0.3096	0.0001 **
**CXCL-10 (pg/mL),** median (IQR)	1056 (215.8–1876)	3266 (702.5–7462)	70.4 (29.7–98)	0.3396	0.0011 **	<0.0001 **

The sex data are expressed in frequency (%); age (years); the cytology, protein, glucose, neopterin, and CXCL-10 measures are expressed in median and interquartile range (IQR) Q1–Q3. Statistical significance of the inflammatory markers was assessed between the groups (A × B, A × C, B × C) using the Kruskal–Wallis test and Dunn’s test of multiple comparisons. Reference values: cell count ≤ 4 cells/mm^3^, protein 15–40 mg/dL, glucose 45–70 mg/dL. ** *p* < 0.05—statistical significance.

## Data Availability

All data used in this study are anonymized and will be shared on request from any qualified investigator.

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
