# Peer review of "Neopterin and CXCL-10 in Cerebrospinal Fluid as Potential Biomarkers of Neuroinvasive Dengue and Chikungunya"

_pathogens, 2021, doi:10.3390/pathogens10121626_

Round 1

Reviewer 1 Report

Pathogens

Manuscript review

Research Article

NEOPTERIN AND CXCL-10 IN CEREBROSPINAL FLUID AS POTENTIAL BIOMARKERS OF NEUROINVASIVE DENGUE AND CHIKUNGUNYA

Authors

Marzia Puccioni-Sohler 1,2,3,*, Samya J. da Silva 1, Luiz C. S. Faria 2, David C. B. I. Cabral 4 and Mauro J. Cabral-Castro 2,5

Abstract

Overall comment: Abstract contains interesting information willing to attract the reader to stay in the article. However, it can be better elaborated, improving the connection between sentences and in specific aspects.

Lines 20-21: The first two statements could be improved. I suggest focusing on the severity of neurological conditions caused by these two pathogens, the mortality rate in this situation, and the difficulty of diagnosis, requiring approaches capable of facilitating access to the sample and predicting the possibility of prognosis for an aggravated clinical outcome.

Introduction

Overall comment: Introduction provides sufficient support for non-active audiences in the specific area. The detailing of the involvement of cytokines and biomarkers contributes to a better understanding of the article. However, I suggest that the levels of these substances in other diseases be addressed.

Are there any reports in the literature about the use or concentrations of serum and other biological fluids as biomarkers?

In this section or at the beginning of Results, it should be included the composition of the study groups.

Results

Overall comment:  The results are approached concisely. However, the writing can be refined, bringing the results obtained in a more impersonal way.

Some points need further explanation, such as: If the levels are similar for CSF proteins and CXCl-10 for the arbovirus and inflammatory groups and both show a strong correlation (Figure 1C and D), as well as there is no difference between the neopterin concentration between groups. What ensures that both biomarker candidates are specific for the detection or prognosis of dengue and Chikungunya neuroinvasive infection?

Lines 96-98: It would be ideal for specifying which group the high-dose individuals belongs.

Figure 2: Figure legends can be improved by including details about each result. I suggest including how the ROC curve was used to determine the concentration of neopterin and CXCL-10.

Figure 3: Figure legends can be improved by including details about each result. I also suggest including the R and p-value for each graph.

Discussion

The discussion reinforces the imminent importance of providing more accurate ways to diagnose and predict progression to neuroinvasive forms of DENV and CHIKV infections. The relationship between the increase in inflammatory markers CXCL-10 and neopterin became clear, mainly due to measurements performed in the inflammatory and non-inflammatory control groups. However, the correlation between the increase in these candidate biomarkers in neuroinvasive infection by DENV and CHIKV is not clear. I suggest including studies that seek to elucidate the pathological mechanism associated with the increase in CXCL-10 and neopterin in this context (or improve the segment from Lines 149 to 161). To what factors do you attribute the similarity between groups A and B?

Lines 134-135: Which biomarkers are found in serological samples? Is there any biomarker common to both samples?

Lines 137-139: Are there reports or studies relating the concentration and severity of the neurological disease developed? It would be interesting data to add to the article, if available.

Given the non-existence of significant difference between the arbovirus group and both controls (including the control of the inflammatory process), I suggest that the discussion should be readjusted to the results obtained.

References

References used for the construction of the theoretical background are recent and consistent with the article's objective.
